# Pan-Lysyl Oxidase Inhibitor PXS-5505 Ameliorates Multiple-Organ Fibrosis by Inhibiting Collagen Crosslinks in Rodent Models of Systemic Sclerosis

**DOI:** 10.3390/ijms23105533

**Published:** 2022-05-16

**Authors:** Yimin Yao, Alison Findlay, Jessica Stolp, Benjamin Rayner, Kjetil Ask, Wolfgang Jarolimek

**Affiliations:** 1Drug Discovery, Pharmaxis Ltd., Sydney, NSW 2086, Australia; yimin.yao@pharmaxis.com.au (Y.Y.); alison.findlay@pharmaxis.com.au (A.F.); jessica.stolp@pharmaxis.com.au (J.S.); 2Children’s Cancer Institute, Lowy Cancer Research Centre, UNSW Sydney, Kensington, Sydney, NSW 2031, Australia; benjamin.rayner@unsw.edu.au; 3Division of Respirology, Department of Medicine, McMaster University, Hamilton, ON L8N 3Z5, Canada; askkj@mcmaster.ca

**Keywords:** scleroderma, lysyl oxidase, fibrosis, small molecule inhibitor

## Abstract

Systemic sclerosis (SSc) is characterised by progressive multiple organ fibrosis leading to morbidity and mortality. Lysyl oxidases play a vital role in the cross-linking of collagens and subsequent build-up of fibrosis in the extracellular matrix. As such, their inhibition provides a novel treatment paradigm for SSc. A novel small molecule pan-lysyl oxidase inhibitor, PXS-5505, currently in clinical development for myelofibrosis treatment was evaluated using in vivo rodent models resembling the fibrotic conditions in SSc. Both lysyl oxidase and lysyl oxidase-like 2 (LOXL2) expression were elevated in the skin and lung of SSc patients. The oral application of PXS-5505 inhibited lysyl oxidase activity in the skin and LOXL2 activity in the lung. PXS-5505 exhibited anti-fibrotic effects in the SSc skin mouse model, reducing dermal thickness and α-smooth muscle actin. Similarly, in the bleomycin-induced mouse lung model, PXS-5505 reduced pulmonary fibrosis toward normal levels, mediated by its ability to normalise collagen/elastin crosslink formation. PXS-5505 also reduced fibrotic extent in models of the ischaemia-reperfusion heart, the unilateral ureteral obstruction kidney, and the CCl4-induced fibrotic liver. PXS-5505 consistently demonstrates potent anti-fibrotic efficacy in multiple models of organ fibrosis relevant to the pathogenesis of SSc, suggesting that it may be efficacious as a novel approach for treating SSc.

## 1. Introduction

Systemic sclerosis (SSc, scleroderma) is an autoimmune disease characterised by progressive development of fibrosis in the skin, lungs, and other internal organs. In recent years, there has been a growing understanding of the pathophysiology of SSc, resulting in the development of anti-inflammatory treatments, however, there are still no approved therapies to effectively ameliorate fibrosis in the target tissues [1]. The key feature of fibrosis in SSc affected organs is an excessive production and accumulation of extracellular matrix (ECM) components resulting from an increase in collagen synthesis, collagen cross-linking and, as a consequence, increased matrix stability and stiffness [2].

Lysyl oxidase (LOX) belongs to a family of five extracellular, copper-dependent enzymes (LOX and lysyl oxidase-like 1-4 [LOXL1-4]) responsible for catalysing the formation of lysine-derived crosslinks in collagen and elastin. In the process of crosslink formation, lysine residues within elastin and collagen can be converted into hydroxylysine residues by lysyl hydroxylase, and undergo oxidative deamination catalysed by lysyl oxidases to generate unstable aldehydes that spontaneously condense to generate intermolecular covalent crosslinks, leading to the formation of immature crosslinks dehydroxylysinorleucine (DHLNL) and hydroxylysinonorleucine (HLNL). The reaction of immature crosslinks with an additional aldehyde results in the formation of stable trimeric, mature crosslinks pyridinoline (PYD) and deoxypyridinoline (DPD) in the ECM. The cross-linked collagens build and stabilise fibrotic tissues and have increased resistance to enzymatic breakdown [3]. LOXL2 is a paralogue of LOX, which is absent in most adult healthy tissues but up-regulated in fibrotic organs, including the lung [4], kidney [5], liver [6], and heart [7]. In contrast, LOX is ubiquitously expressed and is the primary isoform in the skin and connective tissues, and is significantly upregulated in the SSc patients [8].

Lysyl oxidases have been the target of previous therapeutic approaches for SSc. β-aminopropionitrile (BAPN), an irreversible pan-lysyl oxidase inhibitor, has been trialled in SSc patients but discontinued because of its side effects associated with prolonged treatment at high doses [9]. Similarly, inactivation of lysyl oxidases by copper chelation with D-penicillamine reduces scleroderma and improves renal, cardiac, and pulmonary outcome in patients with diffuse cutaneous SSc [10,11]. However, contradictory opinions were raised on the use of D-penicillamine for SSc due to its toxic effects observed at high doses [12].

PXS-5505 is a small molecule, mechanism-based inhibitor of all members of the lysyl oxidase family. In contrast to BAPN, PXS-5505 is highly selective for the target family of enzymes. It is orally bioavailable, safe and well tolerated in healthy human subjects and possesses excellent pharmacokinetic and pharmacodynamic properties (named PXS-LOX_2 in [13]) [14]. In the current, study we sought to confirm tissue expression levels of the lysyl oxidases in SSc patients and preclinical fibrotic disease models, and to demonstrate the anti-fibrotic efficacy of PXS-5505 (via inhibition of collagen crosslinking) in preclinical fibrotic models resembling SSc.

## 2. Results

### 2.1. Elevated Expression of LOX and LOXL2 in SSc Patients

The skin and lung are the two organs most commonly affected by fibrosis in SSc. To determine the level of LOX and LOXL2 in these two organs in SSc, their respective protein expression was assessed by IHC in the biopsies obtained from SSc patients or healthy subjects. Compared to healthy subjects, the amount (percentage) of collagen was more abundant in the SSc lung (12 ± 2% Control vs. 22 ± 4% SSc) and skin (51 ± 2% Control vs. 67 ± 6% SSc) (Figure 1A). In SSc, LOX staining was prominently upregulated in the skin (2.5 ± 0.6% Control vs. 4.5 ± 1.1% SSc) and lung (3.9 ± 1.9% Control vs. 4.5 ± 1.1% SSc) tissues (Figure 1B). The very strong cellular and peri-nuclear presence of LOX in the skin of SSc patients may be related to changes in transcription [15]. Compared to the healthy subject, LOXL2 was also markedly increased in the SSc lung. As expected, the overall LOXL2 expression in the skin is minimal, as it is not the predominate lysyl oxidase isoform normally found in the skin [15] (Figure 1C).

### 2.2. PXS-5505 Attenuates Bleomycin-Induced Skin Fibrosis

The bleomycin mouse model is frequently used for the evaluation of antifibrotic therapies in preclinical studies of SSc [16]. To mimic the severe fibrotic phenotype in SSc, bleomycin was injected subcutaneously every other day for a total duration of 20 days to induce skin fibrosis. In line with the findings in SSc patients, LOX protein expression significantly increased ~2-fold after bleomycin treatment when compared to the control (Figure 2A). Bleomycin administration significantly elevated dermal thickness by ~1.5-fold (Figure 2B), and strongly induced α-SMA protein expression compared to the normal mouse skin (Figure 2C). Daily oral administration of PXS-5505 (15 mg/kg) significantly ameliorated bleomycin induced LOX overexpression and reduced dermal thickness and α-SMA levels (Figure 2A–C).

In order to confirm that lysyl oxidase activity in the skin was blocked by oral gavage of PXS-5505, the activity was measured 24 h after the last dose in vehicle and PXS-5505 treated animals. In the vehicle group, signal over noise was 1.78 ± 0.38 (*n* = 10) while in the PXS-5505 treated group the specific lysyl oxidase signal was almost abolished (1.11 ± 0.08; *n* = 9; *p* < 0.001). Therefore, orally applied PXS-5505 effectively inhibited lysyl oxidase activity in the skin and ameliorated bleomycin-induced skin sclerosis.

### 2.3. PXS-5505 Attenuates Bleomycin-Induced Pulmonary Fibrosis

Due to its severity, pulmonary fibrosis directly contributes to morbidity and mortality in SSc. Bleomycin instilled via oropharyngeal route serves as a rodent model to replicate the key hallmarks of interstitial lung disease in SSc [16,17,18]. In the current study, the lungs of bleomycin-treated mice had elevated levels of LOX and LOXL2 protein when compared to vehicle-treated animals 21 days after bleomycin dosage (Figure 3A), consistent with the elevated LOX and LOXL2 expression found in lungs of the SSc patients. Bleomycin significantly increased lung weight by 1.5-fold (Figure 3B) and leukocyte count in BALF by 6.5-fold (Figure 3C). With lungs of bleomycin-treated animals exhibiting strong fibrosis (Figure 3D), compared to normal mouse lungs. Daily oral administration of PXS-5505 (15 mg/kg) reduced lung weight towards normal levels but did not significantly affect the elevated leukocyte levels. This result was anticipated from a primarily anti-fibrotic drug which does not directly reduce inflammation. Concurringly, lungs from bleomycin-treated animals administered with PXS-5505 had significantly reduced Ashcroft score and the changes to the lung architecture were less severe when compared to bleomycin-treated animals administered with vehicle (Figure 3B–D).

As the primary mechanism of PXS-5505 is the prevention of cross-link formation, the levels of elastin and collagen crosslinks were measured. Hydroxyproline (HYP) levels were also measured to determine the overall collagen content in the lung tissue because of bleomycin and PXS-5505 exposure. As expected from the high Ashcroft score, bleomycin increased the levels of collagen as measured by HYP and all elastin and collagen cross-links measured. Hydroxyproline levels (Figure 4A), elastin cross-links [desmosine (Des) and isodesmosine (Isodes)] were doubled (Figure 4B) while collagen cross-links increased up to 3-fold (Figure 4C). PXS-5505 treatment significantly ameliorated the elevated levels of hydroxyproline, collagen and elastin crosslinks induced by bleomycin, as measured at 3 weeks following bleomycin insult.

Target engagement, i.e., inhibition of lysyl oxidases in the lung, was determined to confirm that the beneficial effects can be associated with the mode of action of the drug. In mouse lungs, the lysyl oxidase activity is too low to be measured by standard Amplex Red type assays. Therefore, we have developed a very sensitive activity-based bioprobe assay which can also be applied for LOXL2 in rodents [19]. In bleomycin-treated animals receiving vehicle, the signal over noise was 2.9 ± 0.2 (*n* = 7) up from 1.5 ± 0.8 in healthy controls (Control vs. BLM + Vehicle BLM, *p* < 0.05) suggesting that the bleomycin-induced increased LOXL2 protein levels (Control: 1450 ± 180 pg/mL vs. BLM: 4310 ± 490 pg/mL, *p* < 0.01) actually translate into enhanced enzymatic activity. Four hours after the last dose of PXS-5505, the signal to noise levels dropped close to baseline 1.4 ± 0.4 (*n* = 7) demonstrating significant LOXL2 inhibition (BLM + Vehicle vs. BLM + PXS-5505, *p* < 0.05), despite the LOXL2 protein level in the BLM + PXS-5505 group (5530 ± 400 pg/mL) was not different to Vehicle BLM group. This study demonstrates that PXS-5505 inhibits activity of lysyl oxidase in the lung, reduces bleomycin-induced increases in collagen, elastin and their cross-links and thereby ameliorates lung fibrosis.

### 2.4. PXS-5505 Attenuates I/R-Induced Cardiac Fibrosis and Improves Function

Cardiac damage develops as a direct consequence of SSc and manifests as myocardial fibrosis, causing decreased heart function [20]. The therapeutic effect of PXS-5505 was evaluated in a rat model of I/R-induced cardiac fibrosis. Four weeks after induction of I/R, *LOXL1* and *LOXL2* mRNA expression was elevated 1.8- and 2-fold, respectively, compared to the sham operated animals, while *LOX*, *LOXL3* and *LOXL4* were unchanged (Figure 5A). Orally dosed PXS-5505 (15 mg/kg) significantly reduced the area of collagen deposition within the left ventricle by ~50%, as assessed by Masson’s trichrome staining or Picro-Sirius Red (Figure 5B), consequently the infarct expansion index also diminished (Figure 5C). The effect of PXS-5505 on markers in the heart that are positively associated with I/R induced cardiac fibrosis were also measured. As expected, I/R significantly increased mRNA expression of several pro-fibrotic markers compared to sham-operated rats, including a significant ~2.5-fold increase in fibronectin and a significant ~3-fold increase in both collagen 1α and collagen 3α, the major collagen isoforms responsible for scar formation following heart attack [21]. Concurrent with the effect of PXS-5505 on cardiac fibrosis, daily oral administration of PXS-5505 decreased the mRNA expression of these and other pro-fibrotic markers, when measured 4 weeks following surgical intervention (Figure 5D).

To study whether these changes in fibrosis and pro-fibrotic biomarkers may translate into functional improvements, heart function was assessed by echocardiography. Compared to the sham-operated cohort, I/R injury caused a significant decrease in left ventricular fractional shortening and ejection fraction, of ~25 and 18%, respectively. PXS-5505 ameliorated this loss of function and fractional shortening and increased ejection fraction towards the basal levels compared to sham-operated rats (Figure 5E).

### 2.5. PXS-5505 Attenuates UUO-Induced Kidney Fibrosis

Interstitial fibrosis has been observed in SSc and the UUO model which causes severe fibrosis can mimic some of this pathology [22]. After ureteral obstruction an acute onset of renal failure and severe hypertension, accompanied with rapid development of fibrosis, occurs which also changes kidney thickness and kidney to body weight ratio in the UUO kidney compared to the contralateral control kidney. As expected from the severity of the model all members of the lysyl oxidase family are upregulated. Two weeks after obstruction, there was a significant ~70-fold increase in *LOX*, *LOXL1* and *LOXL2*, around 250-fold increase in *LOXL3*, while *LOXL4* was moderately (3.8-fold) increased when compared to mRNA expression in sham-operated mice (Figure 6A). PXS-5505 was able to partially reverse the changes caused by UUO. PXS-5505 treatment (10 mg/kg) significantly elevated the kidney thickness and kidney to body weight ratio in the UUO kidneys towards normal, but did not alter these parameters in the contralateral non-affected kidneys (Figure 6B,C). In addition, in mice subjected to UUO, daily oral administration of PXS-5505 significantly decreased the renal tubulointerstital fibrosis by ~27% compared to the vehicle treated group (Figure 6D).

### 2.6. PXS-5505 Attenuates CCl4-Induced Liver Fibrosis

Liver fibrosis can be found in SSc patients [23] and CCl4-induced liver fibrosis is a standard model to induce severe damage. It has been shown that lysyl oxidases are upregulated in CCl4-induced fibrosis [24]. When CCl4 was given twice weekly for 8 weeks, strong fibrosis developed as measured by Picro-Sirius red staining. In CCl4-treated mice that were given PXS-5505 (15 mg/kg), the area of fibrosis was significantly reduced (Figure 7A).

To confirm the mechanism of action, the levels of HYP and collagen crosslinks were determined. Treatment with CCl4 significantly increased HYP, immature (DHLNL and HLNL) and mature (PYD) collagen crosslinks. Daily oral administration of PXS-5505 showed significantly reduced levels of hydroxyproline, HLNL and PYD levels compared to matched vehicle group in the livers of mice subjected to CCl4 (Figure 7B).

## 3. Discussion

The skin and lung are the two most prevalent organs impacted in SSc patients, and this current study provides evidence for the increased expression of LOXL2 in skin and increased LOX in both tissues in SSc patients. Five models of organ fibrosis show the upregulation of lysyl oxidases, indicating their involvement in these pathologies. In this study, oral application of PXS-5505, a novel pan-lysyl oxidase inhibitor currently in clinical development for myelofibrosis (NCT04676529) and hepatocellular carcinoma (NCT05109052), effectively diminished fibrosis in each model and, therefore, represents an exciting new therapeutic approach for the treatment for SSc.

Lysyl oxidases have been considered a promising drug target for more than 60 years. However, the prototype small molecule inhibitor, BAPN, displayed some toxicity at higher doses [9], postulated to be due to a lack of selectivity over other amine oxidase enzymes although promising data were also generated [25]. More recently, a LOXL2 monoclonal antibody failed in several clinical trials [26,27,28], most likely because it was unable to fully block the enzymatic activity in humans [19]. In contrast, this study uses a pan-lysyl oxidase inhibitor, which has successfully completed pre-clinical development, Phase 1 studies, and is currently in Phase 2 for myelofibrosis and hepatocellular carcinoma. In contrast to BAPN, PXS-5505 has an excellent safety profile. When compared to simtuzumab, PXS-5505 effectively and irreversibly blocks enzymatic activity, both in the animal models described in this study as well as in humans [14]. LOXL2 activity in the lung was inhibited by PXS-5505 treatment, demonstrating successful target engagement by the compound in the disease relevant tissue. Although the LOX target engagement in the lung was not directly measured due to low assay sensitivity, it is likely that the LOX activity in the lung was also inhibited by PXS-5505. Significant inhibition of LOX activity in the skin has been achieved.

In SSc, interstitial lung disease accounts for most of the SSc-related mortality attributed to the progressive development of fibrosis in the organ [29]. The current study is the first to show elevated protein levels of both LOX and LOXL2 in the interstitial lung tissue of SSc patients, confirming recently published mRNA data [18]. Concurringly, bronchial and alveolar epithelium of IPF patients show a strong induction of not only LOX but also LOXL2 and LOXL3 gene expression [30]. In the lungs from IPF patients, the LOXL2 isoform has been demonstrated to be crucial for the trans-differentiation of fibroblasts, leading to the development of fibrosis [30]. Higher serum LOXL2 level are also associated with increased risk for IPF progression in patients [31]. A small molecule LOXL2 specific inhibitor has shown antifibrotic efficacy in the bleomycin model [32,33]. In the current study, the antifibrotic efficacy of PXS-5505 was mediated by inhibiting collagen and elastin crosslinks. It is also possible that PXS-5505 inhibited the involvement of LOXL2 as a pro-fibrotic transcription factor at the nuclear level [34]. In fact, LOXL2 inhibition alone may be sufficient to alleviate mild to medium levels of pulmonary fibrosis in animal models and human IPF patients [35]. However, in the fibrotic condition as severe as SSc, where both LOX and LOXL2 are upregulated, pan-lysyl oxidase inhibition using PXS-5505 may be more efficacious by inhibiting all lysyl oxidase isoforms. Interestingly, LOX expression was also downregulated by lysyl oxidase inhibition, likely due to the feedback regulation of LOX synthesis because of reduced tissue stiffness from treatment with PXS-5505. Furthermore, the leukocyte count in bronchoalveolar lavage was not significantly changed as the anti-fibrotic PXS-5505 has no direct impact on inflammation. These data suggest that acute severe forms of lung fibrosis, as seen in SSc and severe IPF, may have several lysyl oxidase isoforms upregulated while more gradually developing fibrosis may be driven by LOXL2 and LOXL3.

The primary isoform of the lysyl oxidase family is LOX and it is found in normal skin [15] and here we show that it is upregulated in SSc patients, similar to previously described data [8]. Although, in the current study, the protein level of LOXL2 was slightly lower in SSc skin, primarily due to its low expression overall, LOXL2 mRNA expression in isolated fibroblasts from SSc patients was markedly elevated [36]. Other isoforms of lysyl oxidase may also contribute to the fibrosis, as LOXL4 expression is elevated in the 3-dimensional cultured human dermal fibroblasts isolated from SSc patients, contributing to increased rigidity, α-SMA expression and collagen crosslinks in the ex vivo environment [37]. Taken together, the upregulation of several lysyl oxidase isoforms in the diseased skin supports the use of pan-lysyl oxidase inhibitor as an anti-fibrotic treatment for SSc. In the bleomycin-induced skin fibrosis model, oral dosing of the pan-lysyl oxidase inhibitor PXS-5505 significantly reduced dermal thickness and α-SMA expression, demonstrating anti-fibrotic efficacy in the skin.

The occurrence of myocardial fibrosis is a consequence of the pathogenetic process in SSc, leading to heart dysfunction, which can be lethal [20,38]. It is well established that elevated LOXL2 is associated with increased level of cardiac fibrosis and deteriorated heart function [39]. This current study has demonstrated the elevated expression of LOXL1 and LOXL2 in the I/R induced fibrotic heart. In contrast, LOX and LOXL3/4 were not upregulated in this model, suggesting a differentiated regulation of lysyl oxidase isoforms in this organ. Pan-lysyl oxidase inhibition by PXS-5505 not only reduced myocardial collagen content following I/R injury, but also downregulated gene expression of the pro-fibrotic markers, likely mediated by a feedback mechanism because of collagen crosslink inhibition that reduced tissue stiffness. Importantly, reduced fibrosis resulted in an improvement in cardiac function suggesting that inhibition of lysyl oxidases may be beneficial to conditions of heart fibrosis. It is noteworthy that while inhibition of LOXL2/3 alone is sufficient to inhibit fibrosis and improve heart function in a mouse permanent carotid artery model [40], in conditions such as SSc, with fibrosis across multiple organs, inhibition of all lysyl oxidases is beneficial due to the potential upregulation of multiple lysyl oxidase isoforms. Therefore, we consider PXS-5505 to be more suitable for severe forms of fibrosis observed in SSc.

Kidney fibrosis is often observed in SSc patients [22], while liver fibrosis is a rare complication of SSc [23]. The injury and dysfunction of these organs could consequently contribute to disease severity in the affected patients. In relatively mild renal fibrosis models such as in diabetic glomerulosclerosis [5] or cyclosporine-induced [41] renal interstitial fibrosis, LOXL2 inhibition alone is sufficient to ameliorate the progress of fibrosis. In contrast to these models, the severe UUO model resembling SSc, where all lysyl oxidase isoforms were elevated, pan-lysyl oxidase inhibition with PXS-5505 as potent anti-fibrotic therapeutic strategy was required and confirming previous data with BAPN [42]. Likewise, while selective LOXL2/3 inhibition can be sufficient to reduce fibrosis in the CCl4 rat model [40], in more severe models where expression of LOX and LOXL2 expression is upregulated, treatment with BAPN has been shown to downregulate LOX expression [24]. Owing to the severity of SSc, it is expected that the pan-lysyl oxidase inhibitor PXS-5505 may produce the desired high level of antifibrotic efficacy which LOXL2 antibody simtuzumab was lacking [19] while having better clinical tolerability compared to BAPN.

## 4. Materials and Methods

### 4.1. Human Tissue Biopsy

The use of human biopsy tissue for the current study was approved by the Hamilton Integrated Research Ethics Board under protocol #0434 and #11120 (Hamilton, ON, Canada). Informed consent was obtained from all subjects involved in the study. Lung tissues were collected retrospectively from the clinical database with two SSc patients with interstitial lung disease being selected. Patient 1 was a 55-year-old female, active smoker (30-years) with disease duration of 1.5 years. Patient 2 was a 69-year-old female, non-smoker, with disease duration of 5 years. Skin biopsies were collected from scleroderma patients during their clinic visits with an informed consent form. Patient 1 was a 63-year-old male with disease duration of 3 years. Patient 2 was a 68-year-old female with disease duration of 6 years. Five control lung samples were obtained from normal looking lung resections of lung cancer patients. Healthy skin tissues were collected from 2 female volunteers (trainees, 20–30 years old). No demographic information was available for the control tissues.

### 4.2. Preclinical Animal Models

PXS-5505 was given as hydrochloride salt but doses are presented as free base.

#### 4.2.1. Bleomycin-Induced Skin Fibrosis in Mouse

The study was performed by PharmaLegacy Laboratories (Pudong, Shanghai, China; Project Code: PL20-0023). Experimental protocols were reviewed and approved by the PharmaLegacy Laboratories Institutional Animal Care and Use Committee (IACUC Protocol Number: PL20-0023-1).

Male *C57BL/6* mice (8- to 9-weeks-old) were randomly assigned to either (1) control (no bleomycin, *n* = 10), (2) bleomycin with vehicle (*n* = 10) or (3) bleomycin with PXS-5505 (*n* = 10) treatment groups. Starting from day 1, mice in groups (2) and (3) received bleomycin (4 U/kg in 100 µL) every other day for 28 days by intra-dermal injection. The mice in group (1) received the same amount of saline as the control. PXS-5505 (15 mg/kg) or vehicle was administered once per day through oral administration starting from Day 4 to Day 28.

Upon termination of the study (2–4 h post last PXS-5505 dose), mice were culled with an intraperitoneal injection of pentobarbital sodium (0.2 mL of a solution of 54.7 mg/mL) followed by cervical dislocation. Skin of the bleomycin treated area was harvested. Skin was frozen immediately in liquid nitrogen and stored at −80 °C for subsequent analysis of crosslinks and target engagement. The remaining was stored in 10% neutral buffered formalin (NBF) for subsequent histological preparation.

#### 4.2.2. Bleomycin-Induced Lung Fibrosis in Mouse

The study was performed by Aragen Bioscience Inc. (Morgan Hill, CA, USA; Project Code: PC-2177-018-7). This study was performed under the animal use protocol: Non-GLP Study: Evaluation of Prophylactic or Therapeutic Efficacy of Test Compounds in Bleomycin Induced Lung Fibrosis Mice. AUP# 17-0331-M-1.

Male *C57/BL6* mice (22–24 g) were randomly assigned to either (1) control (no bleomycin, *n* = 5), (2) bleomycin with vehicle (*n* = 7) or (3) bleomycin with PXS-5505 (*n* = 8) treatment groups. Animals in groups (2) and (3) were administered bleomycin (1.5 U/kg) on Day 0 via the oropharyngeal route, group (1) received saline via the same route. PXS-5505 (15 mg/kg) or vehicle was administered orally, with the first dose given shortly after first bleomycin dose and then daily throughout the entire study (21 days).

At the end of the study (2–4 h post last PXS-5505 dose), mice were culled with an intraperitoneal injection of pentobarbital sodium (0.2 mL of a solution of 54.7 mg/mL) followed by cervical dislocation. Blood was collected in EDTA coated tubes, centrifuged and plasma obtained. Each plasma sample (10 µL) was mixed with 60 µL ACN containing 200 ng/mL tolbutamide and 50 ng/mL propranolol. After the mixture was vortexed for 1 min, it was centrifuged for 10 min at 13,000 rpm (4 ℃). The supernatant was diluted 1:4 with ultrapure water and injected into the LC-MS/MS system (API 4000) to determine the plasma concentrations of PXS-5505.

The lungs were harvested and weighed. The bronchoalveolar lavage fluid was collected by perfusing the lungs twice with 0.5 mL of Hanks Balanced Salt buffer and pooled. The bronchoalveolar lavage fluid was centrifuged at 1000 rpm at 4 °C for 5 min. The resulting cell pellet was suspended in 200 µL of PBS and total leukocytes counted.

The left lung was fixed in 10% NBF for subsequent histopathological preparation. The right lung was immediately frozen in liquid nitrogen and stored at −80 °C for subsequent analysis of crosslinks, immunoblotting, and target engagement.

#### 4.2.3. Ischemia-Reperfusion (I/R)-Induced Myocardial Fibrosis in Rat

This in vivo study was approved by the Sydney Local Health District Animal Welfare Committee, Australia (Protocol Number: 2016-026).

Male *Wistar* rats (400–450 g) were randomly assigned to either (1) sham (*n* = 7), (2) left anterior descending coronary artery ligation (LAD) with vehicle alone (*n* = 13) or (3) LAD with PXS-5505 (*n* = 9) treatment groups. The LAD rats were subjected to transient (30 min) occlusion of the left coronary artery as previously described [43]. The sham animals underwent identical surgery, except no ligation of the LAD was performed. Immediately prior to surgical intervention and subsequently once daily for a period of 4 weeks, the LAD rats were administered either vehicle alone or 15 mg/kg PXS-5505 via oral gavage.

At the end of the study, a transthoracic echocardiogram was performed using a SonoSite Edge II Ultrasound System using a HSL25×/13-6 MHz transducer (Fujifilm Sonosite, Bothell, WA, USA), viewed in M-mode (A) to measure the LV end-systolic diameter and LV end-diastolic diameter. The rats were then euthanised, hearts excised and processed for mRNA extraction, or histological analysis following Masson’s trichrome or Picro-Sirius red staining, with infarct expansion index measured as previously [44].

#### 4.2.4. Unilateral Ureteral Obstruction (UUO)-Induced Kidney Fibrosis in Mouse

The study was performed by Urosphere SAS (Toulouse, France; Project Code: 16-033-D02/18-001). Experimental protocols were reviewed by the CEEA-122 Ethical Committee for Protection of Animals used for Scientific Purposes and approved by the French Ministry for National Education, Higher Education and Research under the number CEEA-122 2014-19.

Male *C57/Bl6J* mice (8- to 9-week-old) were randomly assigned to either (1) sham (*n* = 12), UUO with vehicle (*n* = 11) and UUO with PXS-5505 (*n* = 13) treatment groups. The UUO was performed as previously described [45]. Briefly, mice were anesthetised with a mix of oxygen-isoflurane. A cutaneous incision was performed on the left flank, followed by a muscular incision. The left ureter was exposed, and ligated (6/0 silk). Then, the flank incision was sutured (muscle wall and the cutaneous wall) and disinfected with antimicrobial agent (Vetadine^®^). The same procedure was applied to sham mice, but without ligation of the left ureter. After surgery, animals were returned to their cage. Food and water were given ad libitum. Immediately one-day post-surgical intervention and subsequently once daily for a period of 13 days, the UUO mice were administered either vehicle alone (*n* = 13) or 10 mg/kg PXS-5505 (*n* = 9) via oral gavage.

At the end of the protocol (14 days after UUO), the mice were culled with an intraperitoneal injection of pentobarbital sodium (0.2 mL of a solution of 54.7 mg/mL) followed by cervical dislocation. The right and left kidneys were removed, cleaned of surrounding tissues, rinsed in physiological saline and blotted on a paper. Each kidney (right and left) was weighed, and the length and thickness measured. Kidneys were then cut into two equal parts along a transversal section. One half of each kidney was fixed in 10% NBF solution at room temperature until histological preparation. The other half was snap frozen in liquid nitrogen, then stored at −80 °C for subsequent qPCR analysis.

#### 4.2.5. CCl4-Induced Liver Fibrosis in Mouse

The study was performed by the Pharmalegacy Laboratories (Pudong, Shanghai, China; Project Code: PL18-0311). Experimental protocols were reviewed and approved by the PharmaLegacy Laboratories Institutional Animal Care and Use Committee (IACUC Protocol Number: PL18-0311-1).

Female *BALB/c* mice (7- to 8-week-old, 18–20 g) were randomly assigned to either (1) control (no treatment, *n* = 6), (2) CCl4 with vehicle (*n* = 12) or (3) CCl4 with PXS-5505 (*n* = 12) treatment groups. Animals in groups (2) and (3) were injected intraperitoneally with CCl4 (4 mL/kg of 25% CCl4 in olive oil) twice weekly for a total period of 8 weeks. PXS-5505 (15 mg/kg) or vehicle was administered through oral administration, with the first dose given shortly after first dose of CCl4 and then daily throughout the entire study.

On the day of sacrifice 24 h after the last PXS-5505 treatment, the animals were terminated with CO_2_ and cervical dislocation. The liver was collected. The right caudal and right medium lobes were fixed 10% NBF solution at room temperature and left until histological preparation. The left and left medium lobes were flash frozen in liquid nitrogen, then stored at −80 °C for subsequent crosslinks analysis.

### 4.3. Crosslinks

Collagen and elastin crosslinks in the tissue samples were analysed as previously described [46]. For collagen crosslinks, briefly, 10 mg of freeze-dried tissue samples were reduced with NaBH4. The material then underwent acid hydrolysis in 6M HCl at 105 °C for 24 h. After drying and reconstituting the samples in water, hydroxyproline and crosslinks were extracted from the hydrolysate using an automated solid phase extraction system (Gilson GX-271 ASPECA system). After extraction and drying, hydroxyproline and crosslinks were analysed by UHPLC-ESI-MS/MS on a Thermo Dionex UHPLC and TSQ Endura triple quad mass spectrometer. For elastin crosslinks, 10 mg of freeze-dried samples were processed similarly to the collagen crosslinks sample treatment except for omitting NaBH4 reduction step. Collagen crosslink analysis could not be performed on the skin, heart or kidney tissues due to the challenges in isolating pure fibrotic tissues in sufficient amounts without staining and the low amount of crosslinks present in some tissues.

### 4.4. Target Engagement Assay

#### 4.4.1. Lung LOXL2 Activity and Protein Level

For extraction of LOXL2, mouse lung tissue was snap-frozen with liquid nitrogen and pulverised. Samples were washed thrice by homogenisation (using Turrax T25) with ice-cold phosphate-buffered saline, pH 7.4 with 0.25 mM phenylmethylsulfonyl fluoride (PMSF) and 1 µL/mL bovine aprotinin (Sigma-Aldrich, St. Louis, MO, USA, A3886) as protease inhibitors at 100 µL/mg, centrifuged at 20,000× *g* for 10 min at 4 °C and supernatant discarded. After the final washing step, the pellet was resuspended in RIPA lysis and extraction buffer (Thermo Scientific™ 89900, Waltham, MA, USA) with ratio of buffer volume to tissue weight at 4:1. The homogenate was sonicated in ice-cold water for 3 × 30 s with resting on ice in between and again centrifuged at 20,000× *g* for 10 min at 4 °C. The supernatant was collected and utilised in the SimoaTM/activity-based probe platform for LOXL2 activity and protein level as described previously [19,47].

#### 4.4.2. Skin Lysyl Oxidase Activity

For the detection of lysyl oxidase activity in the skin, epidermal and dermal layers of the skin were dissected, pre-cooled with liquid nitrogen and pulverised. Samples were washed thrice by homogenisation with ice-cold wash buffer (0.15 M NaCl, 50 mM sodium borate, pH 8.0 with 0.25 mM PMSF and 1 µL/mL bovine aprotinin as protease inhibitors) at 100 µL/mg, centrifuged at 10,000× *g* for 10 min at 4 °C and supernatant discarded. After the final washing step, the pellet was resuspended in extraction buffer (6 M urea, 50 mM sodium borate, pH 8.2 with 0.25 mM PMSF and 1 µL/mL aprotinin as protease inhibitors) with ratio of buffer volume to tissue weight at 4:1. After the 3 h incubation at 4 °C, the mixture was diluted 1:2 with assay buffer containing 50 mM sodium borate (pH 8.2), centrifuged at 20,000× *g* for 20 min at 4 °C. The collected supernatant was spiked with pargyline hydrochloride and mofegiline hydrochloride at final concentrations of 0.5 mM and 1 µM, respectively, to inhibit amine oxidases. The samples (25 µL) were then incubated with or without 600 µM BAPN for 30 min at 37 °C and assayed by adding 25 µL reaction mixture (assay buffer containing 50 mM sodium borate, 120 μM Amplex Red, 1.5 U/mL horseradish peroxidase, and 20 mM putrescine dihydrochloride as substrate, pH 8.2). Fluorescence was measured every 2.5 min for 30 min at 37 °C (with excitation and emission at 544 nm and 590 nm, respectively), and the signal (kinetic value without BAPN) to noise (kinetic value with BAPN) ratio indicated the lysyl oxidase activity in the skin tissue.

### 4.5. RNA Isolation and Real-Time PCR Analysis

Total RNA was extracted with the PureLink RNA Mini Kit according to the manufacturer’s instructions (Ambion, Austin, TX, USA), followed by the cDNA synthesis using SuperScript VILO cDNA Synthesis Kit (Life technologies, Carlsbad, CA, USA). The PCR cycling parameters were 15 s × 40 cycles at 95 °C. Glyceraldehyde 3-phosphate dehydrogenase (GAPDH) was used as the house keeping gene in comparisons of gene expression data. Gene expression was measured by the 2^−ΔΔCT^ method using ABI7500 (Applied Biosystems, Waltham, MA, USA) with the ABI TaqMan primer sets as specified in Table 1 below:

### 4.6. Immunoblotting

Lung tissues were homogenized and extracted in RIPA Lysis and Extraction Buffer (Thermo Scientific™ 89900, Waltham, MA, USA) with protease and phosphatase Inhibitor Cocktail added (Thermo Scientific™ 78430, Waltham, MA, USA). The protein concentration was measured using Qubit™ Protein Assay Kit (Thermo Fisher Scientific™ Q33212, Waltham, MA, USA) according to the manufacturer’s instructions. The protein samples were denatured at 70 °C for 10 min and run on NuPAGE^®^ Novex^®^ 4–12% Bis-Tris gel system (Life Technologies NP0321BOX, Carlsbad, CA, USA) for detection of LOX and LOXL2. Gels were electroblotted onto the nitrocellulose membrane (Amersham Pharmacia Biotech, Amersham, UK), and target proteins were captured by the corresponding primary-secondary antibody pairs. Primary antibodies used were LOX (Abcam, Cambridge, UK, Ab31238, 1:200), LOXL2 (R&D, AF2639, 1:200) and vinculin (Sigma-Aldrich, St. Louis, MO, USA, V9264, 1:5000) for loading control. Super-signal West Pico Chemiluminescent Substrate (Thermo Fisher Scientific™ PI-34087, Waltham, MA, USA) was used for signal development and ChemiDoc™ Touch Imaging System (NRNQ1432) was used for imaging.

### 4.7. Histology and Immunohistochemistry

For human tissue biopsies, paraffin embedded sections (4 µm) were dewaxed, and antigen retrieval was performed using Epitope Retrieval 1 (AR9961) for 20 min. Slides were stained using the Bond Polymer Refine Detection kit (DS9800). Rabbit anti-human LOX antibody (Abcam, Cambridge, UK, AB31238) was diluted 1:500 in Power Vision Super Blocker PV6122 to detect human LOX. Goat polyclonal anti-human LOXL2 antibody (R&D Systems, AF2639, 1:500) was used to detect human LOXL2 expression.

For animal tissues, paraffin embedded sections (4 µm) were stained with Picro-Sirius Red (PSR), Masson’s Trichrome (MTC) and Periodic acid–Schiff stains respectively. For each sample, six random non-overlapping fields were captured at 200× magnification using a bright-field microscope (Leica Microsystems, Wetzlar, Germany). The pathological scoring was performed by two independent examiners. For PSR, the area of staining was estimated by Image J software.

Tissues fixed in 10% NBF for 24–48 h were transferred to 70% ethanol and stored at room temperature. Dermis segments were specially fixed against a flat surface in an attempt to control skin puckering. The dehydrated samples were processed and embedded in paraffin. Paraffin embedded tissue sections (3–5 µm) were stained with PSR for the heart, kidney, and liver tissues, and with MTC stains for the lung, skin, and heart tissues. Sectioned samples were processed for antigen release and stained by immunohistochemistry using primary antibodies against LOX (Abcam, Cambridge, UK, AB31238) or α-SMA (Abcam, Cambridge, UK, AB5694).

### 4.8. Analysis

The fibrotic index in bleomycin lung slides stained with MTC was evaluated by the Ashcroft Score as previously described [48]. For each animal, consecutive lung fields were examined in a raster pattern using a 20× objective lens and a 10× ocular lens (200×). A modified Ashcroft Score was recorded for each field. The fibrotic index was calculated as the sum of the modified Ashcroft field scores divided by the number of fields examined.

In the bleomycin skin model, dermal thickness was calculated on three regions from two independent sections that had been previously stained with MTC and measured using Image-J software. Skin slides stained by LOX immunohistochemistry (IHC) were evaluated by light microscopy using the scale scoring system: 1 = Baseline, 2 = Mild, 3 = Moderate and 4 = Severe. Scores were recorded for four fields from each sample and used to calculate a per-animal mean score. The α-SMA IHC was evaluated by an H-score as described [49]. The H-score was determined while blinded using the following equation; H-score = region × (1 + intensity score). Region scoring was determined as follows; 0 = no myofibroblasts, 1 = myofibroblasts only in basal dermis area, 2 = myofibroblasts in basal and lower dermis area, 3 = myofibroblasts in basal and up to middle dermis area, 4 = myofibroblasts across entire dermis. Intensity was determined based on darkness of stain, 0 = no staining, 1 = very pale staining, 2 = pale staining, 3 = medium staining, 4 = dark staining. Five regions from each section were scored with the average taken, two independent sections were then averaged to generate final H-score.

For heart, kidney and liver tissues stained with PSR, the percentage area of staining was estimated by Image-J software.

### 4.9. Statistics

Data were presented as mean ± SEM. Statistics were analysed by Graphpad Prism v8.4 software. The score of LOX immunohistochemistry (Figure 2A), the H-score of α-SMA (Figure 2C) and the fibrotic index values of the Ashcroft Score (Figure 3D) between the indicated treatment groups were compared using the unpaired Mann–Whitney U test. Kidney thickness (Figure 6B) and kidney body to weight ratio (Figure 6C) were analysed using two-way ANOVA with the multiple Bonferoni test comparing the values between treatment groups. All other comparisons were analysed using the unpaired *t*-test. *p* < 0.05 was considered statistically significant.

## 5. Conclusions

The current study demonstrated that both LOX and LOXL2 are elevated in the lung and skin of SSc patients, consistent with higher levels of LOX in the peripheral blood of SSc patients [50]. In addition, other isoforms of the lysyl oxidase family are likely to be upregulated in multiple organs, as demonstrated in the animal models conducted in the current study that resemble SSc fibrotic processes in the heart, kidney, and liver. By inhibiting all isoforms of lysyl oxidases, the novel pan-lysyl oxidase inhibitor PXS-5505 successfully ameliorates aggressive fibrosis progression and may therefore be suitable for treating severe fibrotic diseases such as SSc.

## 6. Patents

Pharmaxis has filed a patent application for the composition and use of haloallylamine derivatives, including PXS-5505, for the treatment of organ fibrosis. Title: Haloallylamine sulfone derivative inhibitors of lysyl oxidases and uses thereof. Publication Number: WO 2020/024017. Publication Date: 2 February 2020. Applicates: Pharmaxis, Ltd.

## Figures and Tables

**Figure 1 ijms-23-05533-f001:**
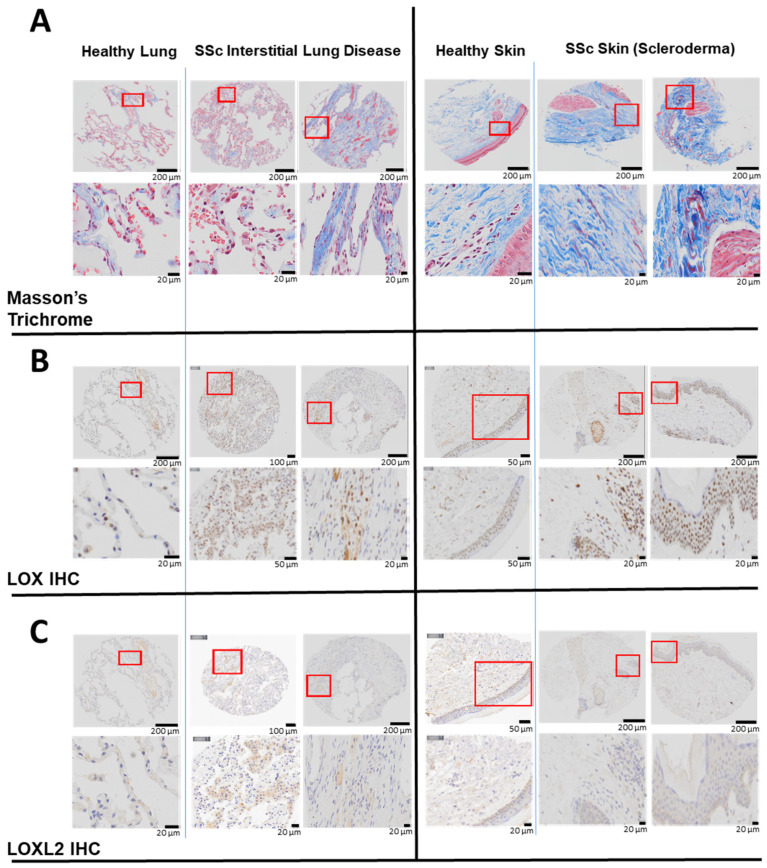
Elevated LOX and LOXL2 and collagen expression in the lung and skin of human patients with SSc. In lung tissues, LOX and LOXL2 was mainly expressed by macrophages and to a lesser extent by the pneumocytes, and endothelial cells of the lung vessels. In skin, LOX and LOXL2 was mostly expressed by the keratinocytes and epithelial cells within the sebaceous glands and the hair follicles. Positive macrophages and fibroblasts can be seen as well. LOX can be seen highly expressed in the nucleus and cytoplasm in comparison to a weaker signal of LOXL2 and more in the cytoplasm than the nucleus. (**A**) Masson’s Trichrome staining revealed higher expression of collagen in the lung biopsies from two patients with interstitial lung disease and SSc compared to a healthy human subject, and higher collagen expression in the skin biopsies from a separate group of two patients with scleroderma compared to normal skin from a healthy subject. (**B**) Immunohistochemistry (IHC) staining using a specific primary antibody against human native LOX revealed higher expression of LOX in the above lung and skin biopsies compared to the respective healthy subjects. (**C**) Immunohistochemistry (IHC) staining using a specific primary antibody against human native LOXL2 revealed higher LOXL2 expression in the lung biopsies from above SSc patients compared to a healthy human subject. LOXL2 expression in the skin is minimal in both the SSc patients and the healthy human subject.

**Figure 2 ijms-23-05533-f002:**
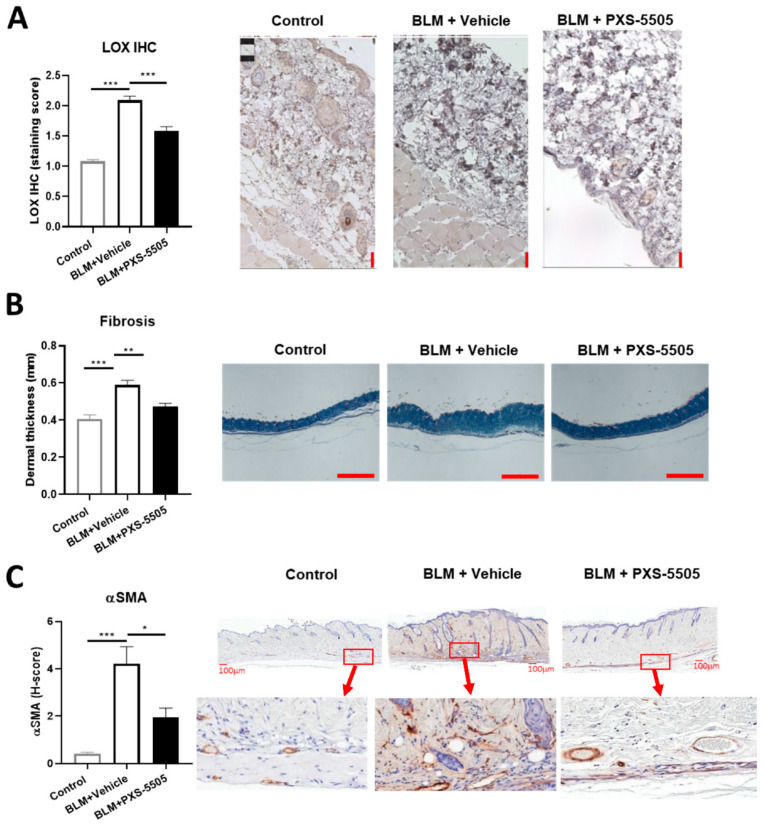
Effect of PXS-5505 treatment on intradermal bleomycin induced skin fibrosis. (**A**) Mice that received intradermal injection of bleomycin showed significantly higher LOX expression level in the skin that was ameliorated by PXS-5505 treatment. Skin slices were immunohistochemically stained against LOX and expression semi-quantified by scoring system. Scale bar = 50 µm. (**B**) Bleomycin treated mice showed significantly higher dermal thickness that was ameliorated by PXS-5505 treatment. Skin tissue slices were stained with Masson’s trichrome. Scale bar = 1000 µm. (**C**) Bleomycin treated mice showed significant higher α-smooth muscle actin (SMA) level, which was ameliorated by PXS-5505 treatment. Skin slices were immunohistochemically stained against α-SMA and expression semi-quantified by H-scoring system. Scale bar = 100 µm and 30× magnification (* *p* < 0.05, ** *p* < 0.01, *** *p* < 0.001 between the indicated groups).

**Figure 3 ijms-23-05533-f003:**
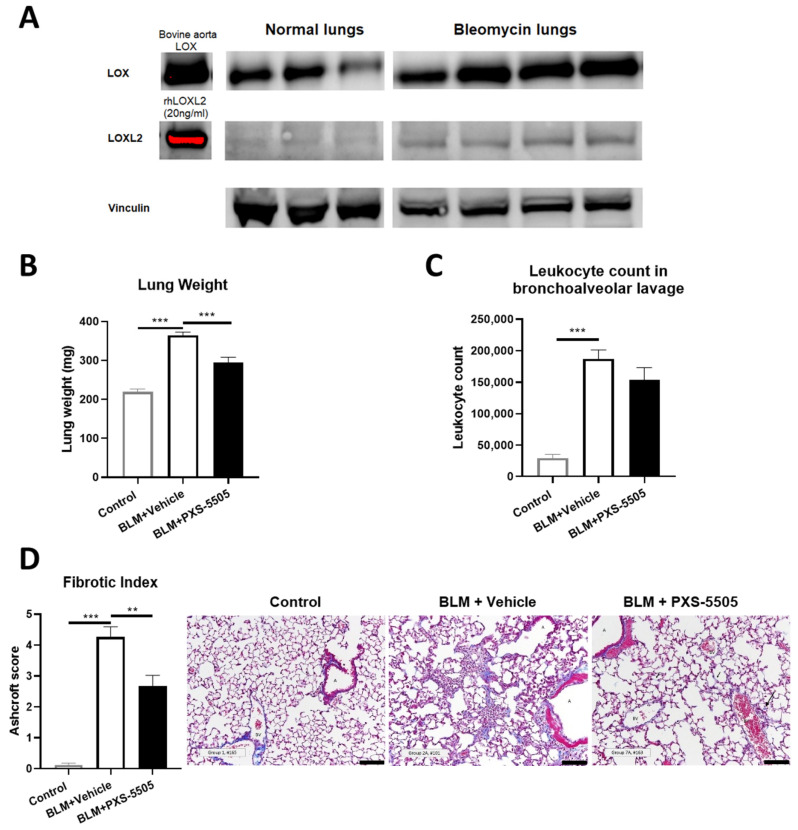
Effect of PXS-5505 treatment on bleomycin induced pulmonary fibrosis in mouse. (**A**) Western blot showing elevated protein expression of LOX and LOXL2 in bleomycin induced fibrotic mouse lungs. (**B**) Elevated gross lung weight in bleomycin treated animals was significantly lowered by PXS-5505 treatment. (**C**) Increased leukocyte count in bronchoalveolar lavage was determined. The PXS-5505 treatment did not significantly change the level. (**D**) Bleomycin treated mice showed significant elevation in the levels of fibrosis that were ameliorated by PXS-5505 treatment. Lung tissue slices were stained with Masson’s trichrome to differentiate fibrotic tissue (blue) and fibrosis semi-quantified by the validated Ashcroft scoring system; A = airway, BV = blood vessel, scale bar = 100 µm (** *p* < 0.01, *** *p* < 0.001 between the indicated groups).

**Figure 4 ijms-23-05533-f004:**
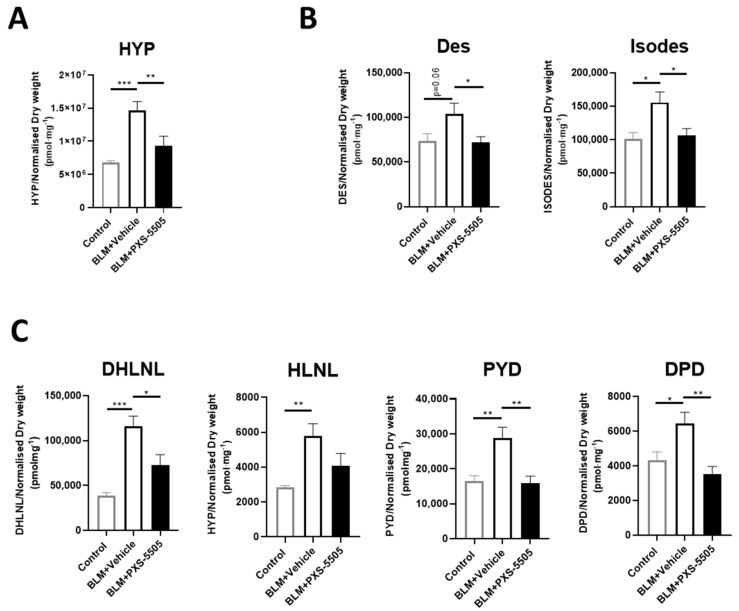
Effect of PXS-5505 treatment on collagen content and crosslinks of elastin and collagen in bleomycin-induced fibrotic mouse lung. Bleomycin elevated the expression of (**A**) hydroxyproline (HYP), (**B**) elastin crosslinks [desmosine (Des) and isodesmosine (Isodes)] and (**C**) immature [dihydroxy-lysinonorleucine (DHLNL) and hydroxylysinonorleucine (HLNL)] and mature [pyridinoline (PYD) and deoxypyridinoline (DPD)] collagen crosslinks in the mouse lungs; PXS-5505 treatment reduced these crosslinking markers induced by bleomycin. Levels were measured using LC-MS/MS methodologies (* *p* < 0.05, ** *p* < 0.01, *** *p* < 0.001 between the indicated groups).

**Figure 5 ijms-23-05533-f005:**
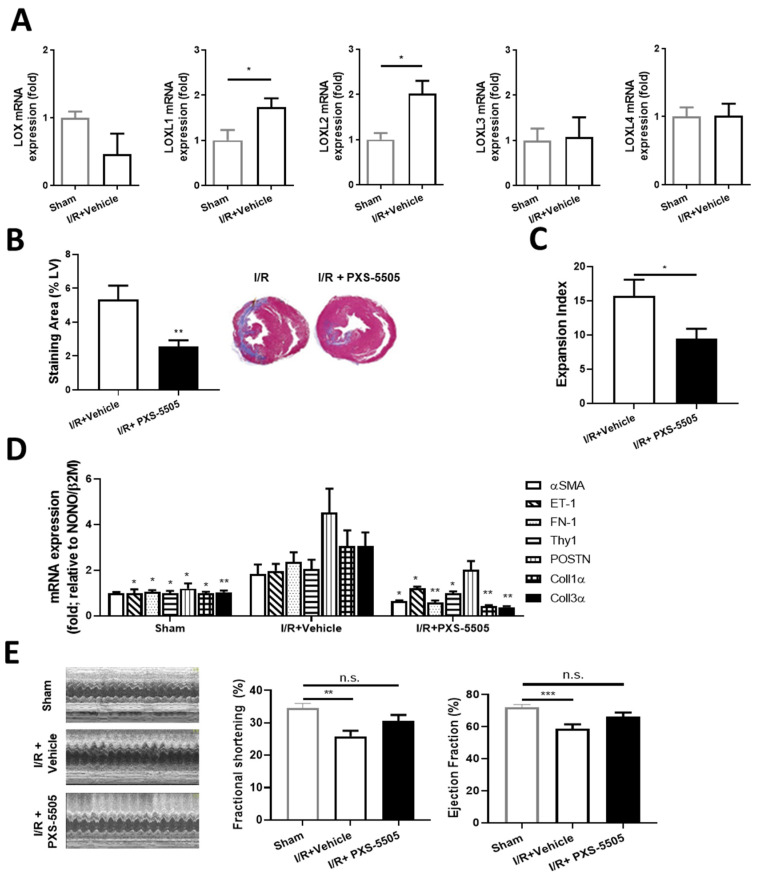
Effect of PXS-5505 treatment on fibrosis and heart function following ischaemia-reperfusion induced myocardial injury. (**A**) mRNA expression showing isoforms *LOXL1* and *LOXL2* were elevated but levels were unchanged for *LOX*, *LOXL3* and *LOXL4* in the I/R induced fibrotic rat hearts at 4 weeks post-surgical intervention. (**B**) PXS-5505 reduced fibrotic area in the rat heart subjected to I/R. Representative images of left ventricle (LV) stained with Masson’s trichrome to differentiate fibrotic tissue (blue) from muscle tissue (purple) with quantification of the staining area expressed as the percentage of the LV. (**C**) PXS-5505 reduced expansion index in the I/R heart. Measurement of expansion index, representing extent of infarct within the LV. (**D**) mRNA expression of fibrosis-related genes in rat hearts at 4 weeks post-surgical intervention were elevated in the group subjected to I/R compared to sham-operated, and returned towards normal in the I/R group with PXS-5505 treatment. The extent of mRNA expression was assessed for rat smooth muscle cell alpha actin (*αSMA*), endothelin 1 (*ET-1*), fibronectin 1 (*FN-1*), Thy1, periostin (*POSTN*), collagen 1 alpha (*Coll1α*) and collagen 3 alpha (*Coll3α*). PXS-5505 improved left ventricular function subjected to I/R. (**E**) Left ventricular function was assessed by echocardiography to measure the LV end-systolic diameter and LV end-diastolic diameter. The mean measurements of three independent recordings performed in triplicate for each rat were used to calculate fractional shortening and ejection fraction. (* *p* < 0.05, ** *p* < 0.01, *** *p* < 0.001 vs. the control I/R cohort or between the indicated groups).

**Figure 6 ijms-23-05533-f006:**
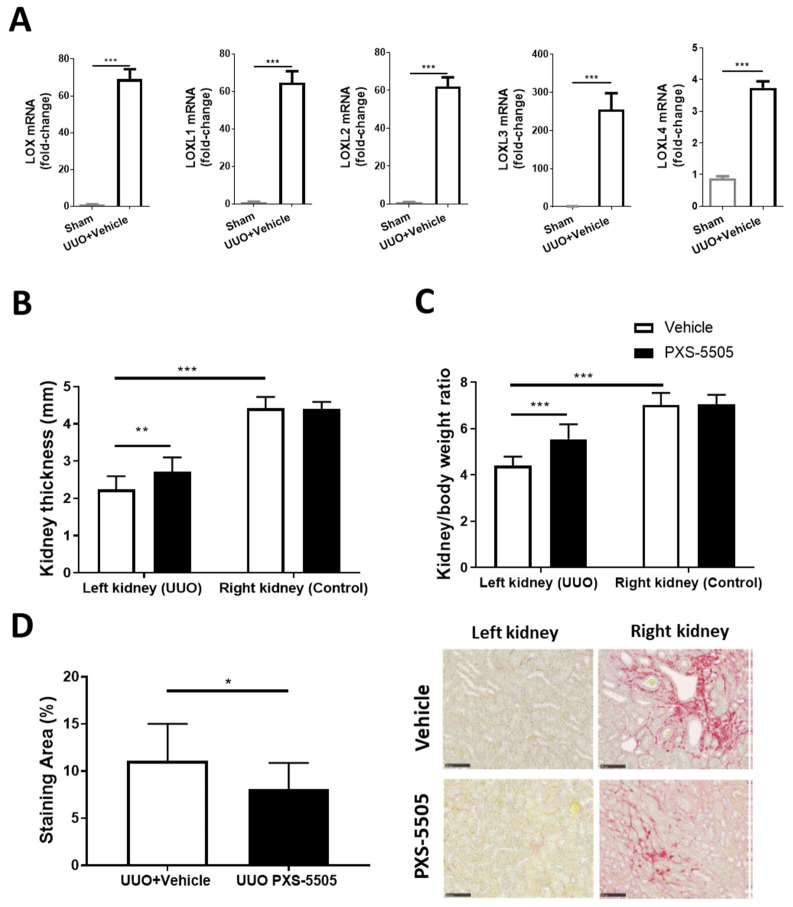
The effect of PXS-5505 treatment on unilateral ureteral obstruction induced kidney fibrosis and LOX over-expression in mouse kidney. (**A**) mRNA expression showing significant elevation of *LOX* and isoforms *LOXL1-4* in the kidneys subjected to UUO compared to the control group that had been subjected to sham surgery. (**B**,**C**) Kidney weight and thickness measured upon sacrifice, being lowered by UUO and returned towards normal values followed by PXS-5505 treatment. (**D**) Representative images of kidney stained with Picro-Sirius red, differentiating collagen (red) from tubulointerstital tissue (yellow) with quantification of the staining area, showing significantly decreased percentage of collagen deposited area in the UUO kidneys in the animals subjected to PXS-5505 treatment compared to the vehicle treated group. Scale bar = 100 µm (* *p* < 0.05, ** *p* < 0.01, *** *p* < 0.001 between the indicated groups).

**Figure 7 ijms-23-05533-f007:**
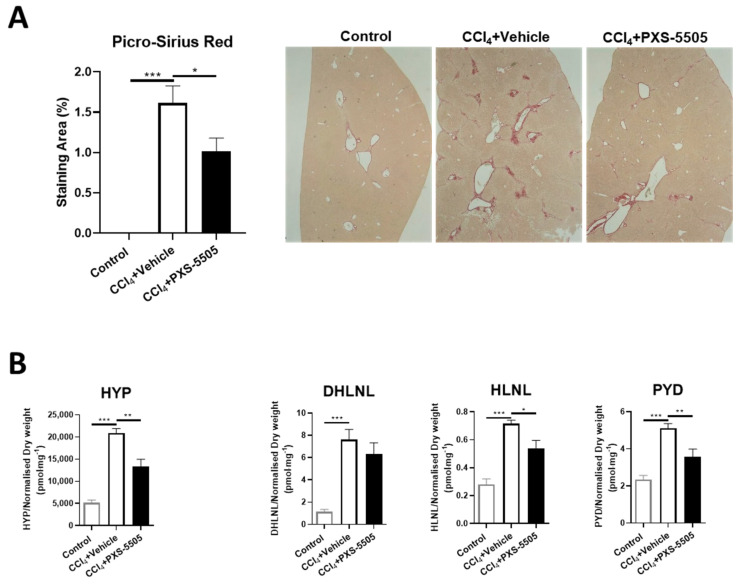
The effect of PXS-5505 treatment on CCl4-induced liver fibrosis and collagen crosslinks in mouse lung. (**A**) Representative images of liver stained with Picro-Sirius red, differentiating collagen (red) from hepatocytes (yellow) with quantification of the staining area, showing significantly elevated percentage of collagen deposited area in the mice administered with CCl4 compared to the normal mice, and the fibrotic area reduced in the CCl4 group subjected to PXS-5505 treatment compared to the vehicle treated group. (**B**) CCl4 administration elevated the expression of hydroxyproline (HYP), immature [dihydroxy-lysinonorleucine (DHLNL) and hydroxylysinonorleucine (HLNL)] and mature [pyridinoline (PYD)] collagen crosslinks in the mouse liver; PXS-5505 treatment ameliorated these elevated markers induced by CCl4. (* *p* < 0.05, ** *p* < 0.01, *** *p* < 0.001 between the indicated groups).

**Table 1 ijms-23-05533-t001:** Information on ABI TaqMan primer sets.

Gene Symbol	Gene	Assay Id.
*LOX*	lysyl oxidase	Rn01491829_m1
*LOXL1*	lysyl oxidase-like 1	Rn01418038_m1
*LOXL2*	lysyl oxidase-like 2	Rn01466080_m1
*LOXL3*	lysyl oxidase-like 3	Rn01765241_m1
*LOXL4*	lysyl oxidase-like 4	Rn01410872_m1
*αSMA*	alpha 2 smooth muscle actin	Rn01759925_g1
*Edn-1*	endothelin 1	Rn00561129_m1
*FN-1*	fibronectin 1	Rn01401499_m1
*Thy1*	Thy-1 cell surface antigen	Rn00664709_g1
*Postn*	periostin	Rn01494625_m1
*Col1α1*	Collagen, type I, alpha 1	Rn01463848_m1
*Col3a1*	Collagen, type III, alpha 1	Rn01437681_m1
*Gapdh*	Glyceraldehyde 3-phosphate dehydrogenase	Mm99999915_g1

## Data Availability

The data presented in this study are available on request from the corresponding author. Some data may not be made available due to privacy or ethical restrictions.

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
