# Peer review of "Pan-Lysyl Oxidase Inhibitor PXS-5505 Ameliorates Multiple-Organ Fibrosis by Inhibiting Collagen Crosslinks in Rodent Models of Systemic Sclerosis"

_ijms, 2022, doi:10.3390/ijms23105533_

Round 1

Reviewer 1 Report

The manuscript "Pan-lysyl oxidase inhibitor PXS-5505 ameliorates multiple-organ fibrosis by inhibiting collagen crosslinks in rodent models of systemyc sclerosis! describes anti-fibrotic effects in different fibrosis models. Results are interesting and the manuscript is well written and designed. However, I have some concerns:

-What is the size of scale bar in the different figures? I can not distingish the number with the size. magnification used should be added in the figure legend.

  • In Fig3A authors show an increase in LOX2 and LOXL2 in bleomycine-treated lungs. What happens with LOX and LOXL2 levels after PXS-5505 treatment?. PXS-5505 reduces lysil oxidase activity in the skin and LOXL2 activity in the lung but authors have not included protein expression levels after the treatment with the inhibitor. Protein exressiion levels of LOX and/or LOXL2 in the different fibrosis models should be analyzed.
  • The lung image in Fig 3D looks fuzzy. Please, impove tue quality of the image.
  • In Fig 6A authors have only analyzed mRNA levels, what happens with protein expression?what happens with PXS-5505 treatment?
  • Images in Fig 7A are dark. please improve tue quality

Author Response

Response to Reviewer 1 Comments

The manuscript "Pan-lysyl oxidase inhibitor PXS-5505 ameliorates multiple-organ fibrosis by inhibiting collagen crosslinks in rodent models of systemyc sclerosis! describes anti-fibrotic effects in different fibrosis models. Results are interesting and the manuscript is well written and designed. However, I have some concerns:

Point 1: What is the size of scale bar in the different figures? I can not distingish the number with the size. magnification used should be added in the figure legend.

Response 1: Scale bars and numbers have been added to the images in figure 1 (page 3), and to the figure legend for figure 2 (page 4), figure 3 (page 5) and figure 6 (page 9). Scale bar is not present within the images of figure 7A as they are an overview of a liver lobe.

Point 2: In Fig3A authors show an increase in LOX2 and LOXL2 in bleomycine-treated lungs. What happens with LOX and LOXL2 levels after PXS-5505 treatment?. PXS-5505 reduces lysil oxidase activity in the skin and LOXL2 activity in the lung but authors have not included protein expression levels after the treatment with the inhibitor. Protein exressiion levels of LOX and/or LOXL2 in the different fibrosis models should be analyzed.

Response 2: Using the SimoaTM/activity-based probe platform (section 4.4.1), we were able to accurately determine the LOXL2 protein concentration in the lung tissue. The values of LOXL2 protein level in the bleomycin lung are now included in section 2.3 (page 6) and methods in page 15, showing LOXL2 protein level was not changed yet as a result of PXS-5505 treatment. In the skin, we used immunohistochemistry observed that LOX expression was downregulated as a result of PXS-5505. We believe this decreased turnover of LOX synthesis was a secondary effect due to softening of extracellular matrix as a result of crosslink reduction following PXS-5505 treatment. Due to technical limitation i.e. small tissue sample and/or low amount of LOX/LOXL2 protein that is below the detection level, we were not able to measure LOX/LOXL2 protein levels in other organs.

Point 3: The lung image in Fig 3D looks fuzzy. Please, impove tue quality of the image.

Response 3: Image quality has been improved in figure 3D (page 5).

Point 4: In Fig 6A authors have only analyzed mRNA levels, what happens with protein expression?what happens with PXS-5505 treatment?

Response 4: UUO is a severe kidney fibrosis model and we show that all forms of lysyl oxidase expression was upregulated, indicating pan-lysyl oxidase inhibition by PXS-5505 is a good strategy for its treatment. As mentioned in response 2, we were not able to measure LOX/LOXL protein levels for the kidney (and liver) tissue due to technical limitation. Although protein expression could not be determined in the current study, there is abundant evidence in the literature showing increased lysyl oxidases expression in the UUO model (Reference 42 and doi.org/10.1016/j.ekir.2020.02.700).

Point 5: Images in Fig 7A are dark. please improve tue quality

Response 5: Image quality has been improved in figure 7A (page 19).

Reviewer 2 Report

In this well written manuscript the authors analyze the effects of the pan-lysyl oxidase inhibitor PXS-5505 on different murine models resembling SSc fibrosis. Here are a few major/minor comments for consideration before manuscript is accepted for publication:

1) In the materials and methods section specify how many healthy tissue biopsies did you collect.

2) Please provide a semiquantitative analysis of LOX and LOXL2 in the lung and skin of SSc patients (figure 1). Moreover, describe in which kind of cells of lung and skin tissues LOX and LOXL2 are expressed, as well as show positive cells at higher magnification.

3) Similarly, in figure 2 show α-SMA positive cells (identifiable as myofibroblasts) at higher magnification.

Author Response

Response to Reviewer 2 Comments

In this well written manuscript the authors analyze the effects of the pan-lysyl oxidase inhibitor PXS-5505 on different murine models resembling SSc fibrosis. Here are a few major/minor comments for consideration before manuscript is accepted for publication:

Point 1: In the materials and methods section specify how many healthy tissue biopsies did you collect.

Response 1: Information on control subjects and healthy tissue biopsies were added to section 4.1 (page 12).

Point 2: Please provide a semiquantitative analysis of LOX and LOXL2 in the lung and skin of SSc patients (figure 1). Moreover, describe in which kind of cells of lung and skin tissues LOX and LOXL2 are expressed, as well as show positive cells at higher magnification.

Response 2: Semiquantitative analysis was added to the Masson’s trichrome collagen and LOX staining, results are now included in section 2.1 (page 2). Due to a low expression of LOXL2 in each tissue (<2%), a quantitative analysis can be largely biased by the background staining and therefore not included in the manuscript. Due to small number of subjects and amount of tissue samples available, statistical analysis could not be performed. A description of cell types that express LOX and LOXL2 is added to the Figure 1 legend (page 3).

Point 3: Similarly, in figure 2 show α-SMA positive cells (identifiable as myofibroblasts) at higher magnification.

Response 3: Figure 2C (page 4) has been edited, showing an overview of a skin section stained with α-SMA and a magnified area with positive stained cells.

Reviewer 3 Report

This is a comprehensive study demonstrating effectiveness of a new lysyl oxidase inhibitor, PXS-5505, in several experimental models of skin, kidney, lung, liver and myocardial fibrosis. A lot of work has been done and the results are convincing. Nevertheless, there are some concerns to be addressed.

  • The title should be modified. “Models of systemic sclerosis” should be replaced by “models of fibrosis”. Models of myocardial, liver and kidney fibrosis cannot be attributed only to SSc because abnormal fibrosis is observed in many other diseases.
  • What is the mechanism through which PXS-5505 inhibits LOX expression at the mRNA and protein levels, not only its enzymatic activity?
  • Were any aspects of liver and kidney function examined after PXS-5505 administration?
  • How many control subjects were used to collect skin samples? Some information about these subjects (age, sex, comorbidities) should be presented.
  • Was there any effect of PXS-5505 on infarct size in the LAD model? Decrease in myocardial fibrosis could result from direct antifibrotic effect or from reduced infarct size.
  • More details about qRT-PCR should be presented including time and temperatures of consecutive cycle phases as well as housekeeping gene(s).
  • Statistical analysis: Mann-Whitney and t tests are designed to compare 2 groups whereas there were 3 groups in most experiments.

Author Response

Response to Reviewer 3 Comments

This is a comprehensive study demonstrating effectiveness of a new lysyl oxidase inhibitor, PXS-5505, in several experimental models of skin, kidney, lung, liver and myocardial fibrosis. A lot of work has been done and the results are convincing. Nevertheless, there are some concerns to be addressed.

Point 1: The title should be modified. “Models of systemic sclerosis” should be replaced by “models of fibrosis”. Models of myocardial, liver and kidney fibrosis cannot be attributed only to SSc because abnormal fibrosis is observed in many other diseases.

Response 1: We understand the rational for this comment. Systemic sclerosis is a multi-organ fibrotic disease and there is not a single animal model that captures its fibrotic features in every organ. The five animal models used in the current study each shows the fibrotic condition of a single organ that closely resembling to the disease status presents in human SSc. In the literature, many studies have also used the bleomycin lung, skin and LAD ligation models to study the pathological features of SSc. We believe the current title is suitable to the content of this paper, we are also happy to change the title according to the reviewer’s comment if the editor feels it is necessary.

Point 2: What is the mechanism through which PXS-5505 inhibits LOX expression at the mRNA and protein levels, not only its enzymatic activity?

Response 2: PXS-5505 inhibits the activity lysyl oxidases. It does not have a direct impact on the protein level of lysyl oxidases (e.g. it does not directly alter the transcription/translation pathway of protein). Saying that, in the skin (Figure 2A, page4) we observed that LOX expression was downregulated as a result of PXS-5505. We believe this decreased turnover of LOX synthesis was a secondary effect due to softening of extracellular matrix as a result of crosslink reduction following PXS-5505 treatment.

Point 3: Were any aspects of liver and kidney function examined after PXS-5505 administration?

Response 3: UUO is a severe kidney disease model and its function is unlikely to be changed within 14 days of treatment. Moreover, the lost kidney function is likely to be compensated by the contralateral normal kidney so the overall renal function as estimated by creatinine clearance would not been change. Single kidney functional study (e.g. inulin clearance via ureter cannulation) neither cannot be performed as the ureter was obstructed as part of the disease model. Liver function was not determined as the results of the current study formed part of a large study focused on antifibrotic effect PXS-5505. On the other hand, our small molecule inhibitor that specifically targets LOXL2, PXS-5153A, that shares structural similarity within the chemical active site with PXS-5505, shown to improve liver function (ALT/AST levels) in the same animal model (ref 40).

Point 4: How many control subjects were used to collect skin samples? Some information about these subjects (age, sex, comorbidities) should be presented.

Response 4: Information on control subjects and healthy tissue biopsies were added to section 4.1 (page 12).

Point 5: Was there any effect of PXS-5505 on infarct size in the LAD model? Decrease in myocardial fibrosis could result from direct antifibrotic effect or from reduced infarct size.

Response 5: Masson’s trichrome staining of fibrosis was used as an indication of infarct size at the 4w time point studied here, but no other direct measurement was used over the course of treatment with PXS-5505. In this respect, the reviewer is correct that the results in this study have not confirmed whether the drug is working to reduce the initial infarct size, or acts to limit fibrosis. Saying that, PXS-5505 is designed as a specific lysyl oxidase inhibitor, the decrease in myocardial fibrosis is likely to result from direct antifibrotic effect.

Point 6: More details about qRT-PCR should be presented including time and temperatures of consecutive cycle phases as well as housekeeping gene(s).

Response 6: Extra information about qPCR was added to the methods (section 4.5, page 15).

Point 7: Statistical analysis: Mann-Whitney and t tests are designed to compare 2 groups whereas there were 3 groups in most experiments.

Response 7: Unpaired t-test was used to run independent comparisons between two groups in most of the figures. In figure 6B and 6C, two-way ANOVA with multiple Bonferoni comparison test was performed to compare between four groups (added to section 4.9, page 13). 

Round 2

Reviewer 1 Report

Some of my concerns have been addressed. However, the magnification used is not indicated in all figures. Please indicate the magnification

Reviewer 2 Report

The authors answered the questions properly.